# MeLLoC: Lossless Compression with High-order Mechanism Learning

**Xinyue Luo**[1]    **Jin Cheng**[1,2]    **Yu Chen**[2*]
[1]School of Mathematical Sciences, Fudan University
[2]School of Mathematics, Shanghai University of Finance and Economics
xinyueluo21@m.fudan.edu.cn   jcheng@fudan.edu.cn   yuchen@sufe.edu.cn

## Abstract

Lossless compression of large-scale scientific floating-point data is critical yet challenging due to the presence of noise and high-order information that arises from model truncation and discretization errors. Existing entropy coding techniques fail to effectively leverage the mechanisms underlying the data generation process. This paper introduces MeLLoC(Mechanism Learning for Lossless Compression), a novel approach that combines high-order mechanism learning with classical encoding to enhance lossless compression for scientific data. The key idea is to treat the data as discrete samples from an underlying physical field described by differential equations, and solve an inverse problem to identify the governing equation coefficients exhibiting more compressible numeric representations. Periodic extension techniques are employed to accelerate the decompression. Through extensive experiments on various scientific datasets, MeLLoC consistently outperforms state-of-the-art lossless compressors while offering compelling trade-offs between compression ratios and computational costs. This work opens up new avenues for exploiting domain knowledge and high-order information to improve data compression in scientific computing.

## 1   Introduction

In the modern era, vast amounts of floating-point data are generated from large-scale simulations and experiments, often reaching petabyte levels. Efficient compression of data is essential for reducing storage costs and facilitating data transfer and analysis. Consequently, the compression of scientific data has gained increasing attention.

Lossless compression aims to reduce the size of data while allowing perfect reconstruction of the original data from its compressed representation. Classical lossless compression algorithms like Gzip and Bzip2 employ entropy encoding techniques like Huffman coding and arithmetic coding to exploit statistical redundancies in the data[1, 2]. While effective for general-purpose data, these methods typically underperform when applied to scientific floating-point datasets, as they fail to capture the distinct characteristics of such data. In contrast, specialized compression algorithms like FPZIP[3], ZFP[4], zstd[5] and ALP[6] are designed to handle floating-point data more effectively by employing predictive coding or customized entropy coders that account for the spatial correlations and value distributions present in scientific datasets.

A common challenge in scientific data compression is the presence of high-order information and randomness, or noise, which arises from roundoff errors during simulation, as well as model truncation and discretization errors. This noise, termed as "false precision" by Zender[7], can significantly lower the compression ratio. The reason is that lossless compression algorithms do not distinguish the informative mechanism from the non-informative noise in data, thus leaving the level of meaningful precision unassessed and failing to exploit the high-order information in the data. By leaving the

38th Conference on Neural Information Processing Systems (NeurIPS 2024).

noisy components, the data is more likely to exhibit detectable patterns, indicating that their size can be effectively reduced by a compressor[8].

To address this issue, Klöwer et al. propose a method of truncation based on evaluating the bitwise dependence between adjacent grid points [9]. However, denoising by truncation on bit digits may not be an optimal approach in practice due to the non-uniform distribution and varying levels of noise in scientific data across different spatial and temporal contexts, which risks confounding the scientific insights gleaned from the data. A recent work by Luo et al. [10] considers locally characterizing the data's spatial relationship with linear differential equations, then separating the noise instead of using truncation. While this method offers a more nuanced approach to noise handling, it's important to note that both these techniques still constitute forms of lossy compression.

Scientific data typically conforms to well-defined model mechanisms, often characterized using differential equations[11]. According to the theory of well-posedness for differential equations, the solutions to the differential equations are determined by the initial and boundary conditions, as well as the source terms[12]. This suggests that the entire physical field could be represented by a smaller, compressed set of values that describe these mechanisms. However, most existing lossless compression techniques do not exploit this inherent structure and high-order information in scientific floating-point data.

In this paper, we aim to further enhance the effectiveness of lossless compression by learning the inherent mechanisms underlying scientific data, building on previous near-lossless work[10]. The key idea is to treat the data as samples from a discretized physical field and solve an inverse problem for the governing differential equations to obtain the source terms, which exhibit a more compressible numeric distribution.

Our main results can be summarized as follows:

1. We propose MeLLoC (Mechanism Learning for Lossless Compression), a novel approach that combines high-order mechanism learning with classical encoding algorithms to achieve efficient lossless compression of scientific floating-point data.

2. We implement periodic extension techniques to accelerate the solution of the large linear systems arising in the parameter identification problem.

3. Through comprehensive numerical experiments, we demonstrate that MeLLoC consistently outperforms the existing methods while offering compelling trade-offs between compression ratios and computational requirements.

## 2 Background

**Partial Differential Equations behind Scientific Data.** Partial Differential Equations (PDEs) are fundamental in modeling complex physical phenomena across various scientific disciplines. They provide a mathematical framework for describing spatiotemporal changes in physical quantities. In oceanography and climate science, PDEs are widely used to model phenomena such as heat transfer, fluid dynamics, and wave propagation. Specifically in oceanography, these equations model ocean circulation, temperature distribution, and salinity patterns, often involving multiple variables and intricate interactions. While the specific equations can vary depending on the phenomenon being studied, they generally capture the fundamental physical principles governing the system's behavior.

The complexity of these PDEs and their numerical solutions often results in large-scale floating-point datasets, which are the focus of our compression efforts. Understanding the underlying mathematical structure of these datasets is crucial for developing effective compression techniques that can preserve important physical features while reducing data volume.

**Mechanism Learning of Scientific data** Mechanism learning, or data-driven discovery of governing equations, has gained significant attention in scientific computing. This approach aims to uncover the underlying physical laws from observational or simulation data. Key techniques in this area include Sparse Identification of Nonlinear Dynamics(SINDy)[13] ,Physics-Informed Neural Networks(PINNs)[14] . These methods provide a foundation for identifying compact representations of complex systems, which can potentially be applied for data compression. However, SINDy requires temporal evolution data to learn the underlying dynamics, while PINNs typically need large datasets for training, making them less suitable for case-by-case data compression scenarios.

**Entropy Encoding.** Entropy encoding is a fundamental lossless data compression technique that assigns shorter codes to frequently occurring symbols and longer codes to rare ones, based on the statistical properties of the data. The encoding scheme is designed to minimize the overall bit length of the encoded message. Shannon's source coding theorem[15] provides the theoretical basis, suggesting that the optimal code length for a symbol $x$ is $-\log \mathcal{D}(x)$, where the expected code length is bounded by the entropy of the source distribution $\mathcal{D}$.

Finite State Entropy (FSE)[16] is a notable entropy encoding method that offers competitive compression ratios and fast encoding and decoding speeds. FSE constructs a finite state machine to model the probability of symbols, adapting to the changing statistics of the input data. This adaptability makes FSE particularly effective for compressing data with non-uniform or unknown distributions.

# 3 Methodology

## 3.1 Overview of the MeLLoC Framework

Building upon PDE theory and mechanism learning concepts discussed earlier, we develop our compression methodology specifically for scientific data. We demonstrate our approach using two-dimensional (2D) scenarios, though the methodology can be extended to higher dimensions. Our method takes advantage of the inherent mathematical structures present in scientific data to achieve more effective compression.

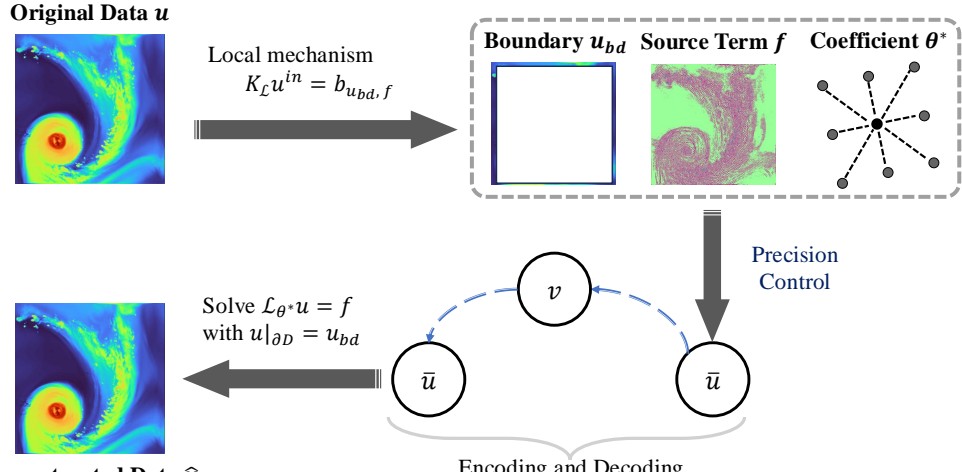

Figure 1: Overview of the proposed compression architecture.

We first exemplify the idea through the Laplace equation, a canonical PDE representing steady-state conditions across various fields such as temperature distribution and electric potential

$$\Delta u = 0.$$

When the data represents the numerical solution of the aforementioned equation, derived from a second-order central difference scheme applied to a grid with uniform spacing, it complies with the following equation

$$u_{i,j} = \frac{1}{4}(u_{i,j-1} + u_{i,j+1} + u_{i-1,j} + u_{i+1,j}).$$

This implies that the value at any central point is calculable based on the values of its immediate neighboring points. Furthermore, in this scenario, the data is entirely determined by its boundary values, suggesting that only the boundary data need be stored to fully reconstruct the original dataset.

In practice, data may exhibit non-homogeneous conditions, described by $\Delta u = f$, where $f$ denotes the source term. In such cases, both the boundary values and the source $f$ need to be stored. Compression remains viable, provided the information entropy of $f$ is subordinate to that of $u$, a condition often satisfied when data exhibits elliptic properties or when the source term is of low magnitude and sparse.

To address data governed by diverse mechanisms, our model is parameterized and calibrated through learning from the data. We assume that a set of two-dimensional data satisfies a difference equation $\mathcal{L}u_{i,j} = f_{i,j}$, where $\mathcal{L}$ represents the difference operator. Globally, a linear system for the interior unknowns $u^{in}$ can be constructed as follows

$$K_{\mathcal{L}}u^{in} = b_{u^{bd},f}. \tag{1}$$

Here, $K_{\mathcal{L}}$ is the stiffness matrix corresponding to the difference operator, while $b_{u^{bd},f}$ comprises boundary data and source term data and represents the 'data support' which refers to the data needed for the well-posedness of the equation.

Assume that the scientific data can be locally approximated by a homogeneous second-order linear PDE with constant coefficients. In this context, our goal is to learn the difference operator $\mathcal{L}$ by minimizing the source term $|f|$, thereby achieving maximal sparsity and reduced information entropy. Consequently, only the boundary data $u^{bd}$ and the sparse source term $f$ are stored. During decompression, the data can be reconstructed using the matrix $K_{\mathcal{L}}$. The proposed method is not limited to specific difference formats and can effectively handle variable coefficient equations.

To facilitate the compression, the difference operator $\mathcal{L}$ is formulated in a compact 9-point form and parameterized by $\theta = \{C_i\}_{i=1}^9$. The coefficient template for the combination of neighboring points is depicted in Figure 2(a), illustrating how local representation translates to global connectivity among data points, as shown in Figure 2(b). Figure 2(c) provides some typical templates for differential equations.

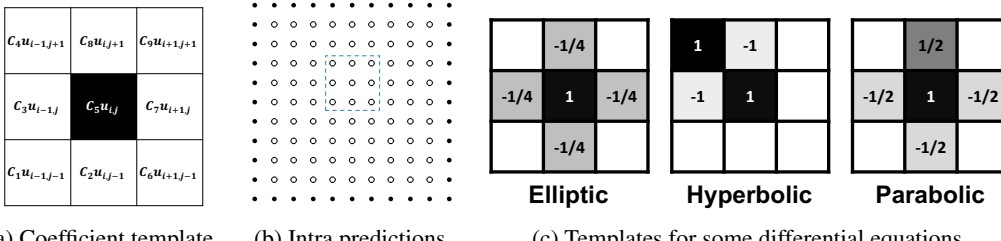

(a) Coefficient template.  (b) Intra predictions.  (c) Templates for some differential equations.

Figure 2: Local representation notations.

In the compression process, we need to determine the parameters $\theta = \{C_i\}_{i=1}^9$ such that

$$C_1 u_{i-1,j-1} + \cdots + C_9 u_{i+1,j+1} = f_{i,j},$$

and minimize the magnitude of source term $f_{i,j}$. Therefore, learning the operator (equivalently, the corresponding stiffness matrix) transforms into solving an optimization problem formulated as

$$\underset{\theta}{\mathrm{argmin}}\, F\left(\theta; u_d^{bd}, u_d^{in}\right),$$

where, by normalizing $\theta$ with $C_5 = -1$, the loss function is defined by

$$F = \|\mathcal{L}_\theta u_d\|_2^2 = \sum_{i,j} \left(C_1 u_{i-1,j-1} + \cdots + C_9 u_{i+1,j+1} - u_{i,j}\right)^2.$$

In applications, to reduce computational cost or highlight local features, a strategic approach involves selecting only a localized subset of the data to learn the difference operator $\mathcal{L}_\theta$. During decompression, we obtain $u^{in}$ by solving equation $K_{\mathcal{L}}u^{in} = b_{u^{bd},f}$.

Our approach is particularly suitable for compressing numerical solutions of differential equations. In addition, it performs well for data with explicit physical mechanisms, such as those encountered in atmospheric and oceanic observational datasets or within the industrial sector. The learned nine-point coefficient template $\mathcal{L}_\theta$ (the difference operator) corresponds to the mechanism described by a second-order differential equation and is generally applicable. Furthermore, the stability of both the compression and decompression processes is assured by the well-posedness of the difference equation.

## 3.2 Model identification and its well-posedness

Consider the underlying model of the data. For $u \in C^4(\Omega)$, $\Omega \subset \mathbb{R}^2$, the nine-point difference template is related to the second order linear differential operator as

$$\sum_{k,l=-1}^{1} C_{k,l} u(x+kh, y+lh) = [(c_3 + c_7 + c_8)h^2\partial_{xx}^2 + (c_4 + c_7 + c_8)h^2\partial_{yy}^2$$

$$+ 2(c_7 - c_8)h^2\partial_{xy}^2 + 2(c_1 + c_5 - c_6)h\partial_x + 2(c_2 + c_5 + c_6)h\partial_y$$

$$+ c_9]u(x,y) + o(h^2).$$

Here $C_{k,l}$ are the coefficients in Figure 2(a) and the subscripts $k, l$ represent the relative position to the data point $(x, y)$. The relationship between $C_{k,l}$ and $c_n$ can be expressed as

$$\mathbf{C} = c_1\mathbf{A}_1 + c_2\mathbf{A}_2 + c_3\mathbf{A}_3 + c_4\mathbf{A}_4 + c_5\mathbf{A}_5 + c_6\mathbf{A}_6 + c_7\mathbf{A}_7 + c_8\mathbf{A}_8 + c_9\mathbf{A}_9.$$

where $\mathbf{C}$ is the matrix of $C_{k,l}$, $\mathbf{A}_n$ are basis matrices, and $c_n$ are corresponding coefficients. The matrices $\{\mathbf{A}_n\}_{n=1}^9$ are defined as

$$\mathbf{A}_1 = \begin{bmatrix} 0 & 0 & 0 \\ -1 & 0 & 1 \\ 0 & 0 & 0 \end{bmatrix}, \mathbf{A}_2 = \begin{bmatrix} 0 & 1 & 0 \\ 0 & 0 & 0 \\ 0 & -1 & 0 \end{bmatrix}, \mathbf{A}_3 = \begin{bmatrix} 0 & 0 & 0 \\ 1 & -2 & 1 \\ 0 & 0 & 0 \end{bmatrix},$$

$$\mathbf{A}_4 = \begin{bmatrix} 0 & 1 & 0 \\ 0 & -2 & 0 \\ 0 & 1 & 0 \end{bmatrix}, \mathbf{A}_5 = \begin{bmatrix} 0 & 0 & 1 \\ 0 & 0 & 0 \\ -1 & 0 & 0 \end{bmatrix}, \mathbf{A}_6 = \begin{bmatrix} 1 & 0 & 0 \\ 0 & 0 & 0 \\ 0 & 0 & -1 \end{bmatrix},$$

$$\mathbf{A}_7 = \begin{bmatrix} 0 & 0 & 1 \\ 0 & -2 & 0 \\ 1 & 0 & 0 \end{bmatrix}, \mathbf{A}_8 = \begin{bmatrix} 1 & 0 & 0 \\ 0 & -2 & 0 \\ 0 & 0 & 1 \end{bmatrix}, \mathbf{A}_9 = \begin{bmatrix} 0 & 0 & 0 \\ 0 & 1 & 0 \\ 0 & 0 & 0 \end{bmatrix}.$$

Based on this representation, encoding $u$ becomes encoding the sparser high-order term $o(h^2)$, i.e., the source term $f$. Therefore, the optimization objective is to obtain a minimized high-order term, which can be mathematically expressed as

$$\{C_{k,l}\}^* = \underset{C_{k,l}}{\operatorname{argmin}} F(C_{k,l}; u) = \underset{C_{k,l}}{\operatorname{argmin}} \sum_{i,j} \left( \sum_{k,l=-1}^{1} C_{k,l} u(i+kh, j+lh) \right)^2.$$

Once the template $\theta := C_{k,l}$ is learned, one can calculate the coefficients of the differential operator, allowing to classify the mechanism as elliptic, parabolic, or hyperbolic due to the reversibility of $C_{k,l}$ and $c_n$, which provides interpretability of the model.

Next, we discuss the well-posedness of the model identification problem (compression process), while that of the decompression process will be discussed in Section 3.5. The above minimization problem is equivalent to solving the least square problem

$$Ac = 0,$$

where $c = [C_1, \cdots, C_9]^T$, $A \in \mathbb{R}^{N \times 9}$, $N$ is the number of data points in domain $D$, $\mathcal{P} : D \to \mathbb{R}$, and $k = \mathcal{P}(i, j)$, $(i, j) \in D$ is the index after rearranging the data into 1-dimensional vector, with $(i, j) = \mathcal{P}^{-1}(k)$, $k = 1, \cdots, N$, as its inverse mapping.

$$A_{k,\cdot} = [u_{i-1,j-1}, \cdots, u_{i+1,j+1}], \quad (i, j) = \mathcal{P}^{-1}(k).$$

To find non-trivial solutions is to obtain the null space (kernel) of $B = (A^T A)$. If $B$ is full-rank, there is only trivial solution, while otherwise, there is no uniqueness. To address this issue, we set the coefficient of $u_{i,j}$ to -1 (i.e. set $C_5$ to -1 in Figure 2(a)), and fix the template size to 8. The problem then becomes

$$\tilde{A}\tilde{c} = b,$$

where $\tilde{c} \in \mathbb{R}^8$, $\tilde{A} \in \mathbb{R}^{N \times 8}$, and $b = [u_{\mathcal{P}^{-1}(1)}, \cdots, u_{\mathcal{P}^{-1}(N)}]^T \in \mathbb{R}^N$. The problem has a unique least squares solution $\tilde{c} = \tilde{A}^\dagger b$, provided the data are not all degenerated. We assemble $\tilde{A}$ and directly solve the pseudo inverse, which serves as a fast solver for the compression process.

### 3.3 Data Composition

To illustrate the method's capacity for lossless compression, consider the decomposition of data $u$ predicated upon linear superposition

$$u = \mathcal{L}^{-1}f + u_0 + u_{err}.$$

This equation stratifies the dataset into three fundamental components:

- $\mathcal{L}^{-1}f$, representing $G * f$, with $G$ being the Green's function for the domain. This part corresponds to solution to the non-homogeneous equation with homogeneous boundary conditions, determined by the source $f$ (2D).

- $u_0(= \frac{\partial G}{\partial \nu} * u^{bd})$ denotes the solution to the homogeneous equation is determined only by the boundary data (1D).

- $u_{err}$ denotes the residual part (2D).

The precision control discussed in the following enables us to optimize the source term and retaining sufficient significant figures based on data precision to nullify $u_{err}$, thereby yielding $u = \mathcal{L}^{-1}f + u_0$. This decomposition allows us to store only the boundary data $u^{bd}$ and the sparse source term $f$, achieving compression while preserving the essential information content of the original dataset.

### 3.4 Precision Control

Suppose the data $u$ possesses a precision of $m$ bits, expressed in decimal form, with a binary equivalent exhibiting similar properties. This precision implies an accuracy to the order of $10^{-m}$, as depicted in Figure 3(a). By analogy, the precision of the model coefficients $C_i$ is quantified as $n$ bits, corresponding to $10^{-n}$. Consequently, when the source term $f$ is computed via $f = \mathcal{L}(u) = \sum_{i=1}^{9} C_i u_i$, the precision of $f$ is up to $10^{-m} \times 10^{-n} = 10^{-(m+n)}$.

The data can be losslessly recovered or decompressed once the error from solving the linear system $K_{\mathcal{L}} u^{in} = b_{u^{bd},f}$ is less than $10^{-(m+n)}$, which is feasible with proper $n$. As $n$ increases, the admissible set for $C_i$ enlarges, contributing to a lower absolute value for $f$ but higher precision. The optimal compression ratio is reached when significant digits of $f$ are minimized. We optimize $n$ for coefficients $C_i$ by starting with high precision and gradually reducing it while monitoring reconstruction error and compression ratio. After several calibrations, $n$ can be fixed for the remaining dataset if the compression ratio for subsequent batches shows no significant fluctuation.

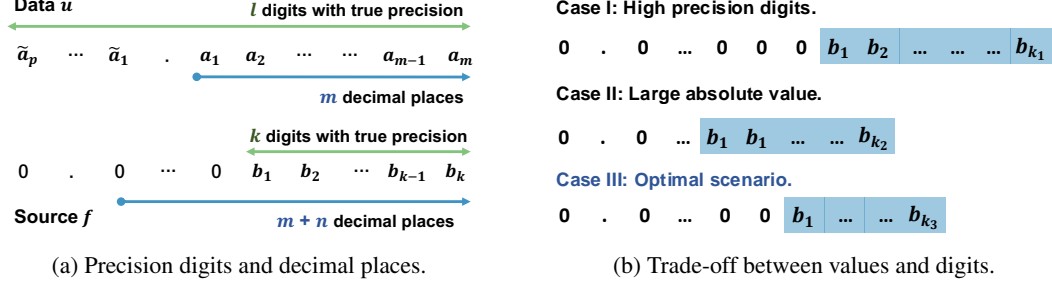

(a) Precision digits and decimal places.    (b) Trade-off between values and digits.

Figure 3: Illustration of precision control.

This approach considers different scenarios to balance significant digits and value magnitudes, as illustrated in Figure 3(b). The trade-off between fewer effective digits of $f$ compared to $u$ contributes to compression, while increasing precision in coefficient truncation expands the solution space, potentially decreasing the absolute value of the optimized source term but increasing storage requirements. Our method, MeLLoC, optimizes compression efficiency, balancing perfect reconstruction with computational feasibility. The precision control is adaptive and can be tailored to different scientific datasets, ensuring that the compression process is optimized for various types of atmospheric and climate model data.

## 3.5 Fast Fourier-based Solver

The traditional approach to solving systems of equations (1) can be computationally intensive, particularly when dealing with large datasets. To address this issue, we propose a Fourier-based solver that accelerates computation while preserving data integrity.

We first extend the discrete field data periodically and expand it using a 2D Fourier series

$$u(m,n) = \sum_{k=1}^{N_2} \sum_{j=1}^{N_1} \frac{1}{N_1 N_2} \hat{u}_{jk} e^{\frac{2\pi i}{N_1}(j-1)(m-1)} e^{\frac{2\pi i}{N_2}(k-1)(n-1)}$$

where $i = \sqrt{-1}$ represents the imaginary unit, and $\hat{u}_{j,k}$ represents the Fourier coefficients. It is essential that the double-layer boundary condition enables periodic extension for the source field $f$. By substituting the Fourier series into the difference equation, we obtain a system relating these coefficients

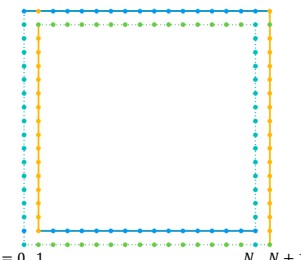

Figure 4: Schematic representation of periodic continuation.

$$\begin{cases} \hat{u}_{j,k} = \frac{\hat{f}_{j,k}}{B_{j,k}}, & B_{j,k} \neq 0 \\ \hat{u}_{j,k} = 0, & B_{j,k} = 0 \end{cases} \quad \text{, for } j, k = 1, \cdots, N_1 \text{ and } N_2. \tag{2}$$

Here, $\hat{f}_{j,k}$ represents the Fourier coefficients of $f$. $B_{j,k}$ is a coefficient involving the difference operator and the Fourier basis functions. The specific form of $B_{j,k}$ depends on the discretization scheme used in the original differential equation[17]. In general cases, the double-layer boundary values are necessary to determine the base frequency Fourier coefficients, ensuring the well-posedness. For example, when $n = 1$,

$$u(m,1) = \sum_{k=1}^{N_2} \sum_{j=1}^{N_1} \frac{1}{N_1 N_2} \hat{u}_{jk} e^{\frac{2\pi i}{N_1}(j-1)(m-1)} = \frac{1}{N_1} \sum_{j=1}^{N_1} \left( \frac{1}{N_2} \sum_{k=1}^{N_2} \hat{u}_{jk} \right) e^{\frac{2\pi i}{N_1}(j-1)(m-1)}.$$

By applying DFT to $u(m,1)$, we obtain $\tilde{u}_j$ satisfying $u(m,1) = \frac{1}{N_1} \sum_{j=1}^{N_1} \tilde{u}_j e^{\frac{2\pi}{N_1}(j-1)(m-1)}$. Utilizing

the orthogonality, we can derive $\tilde{u}_j = \frac{1}{N_2} \sum_{k=1}^{N_2} \hat{u}_{jk}$ and therefore $\hat{u}_{j1} = N_2 \tilde{u}_j - \sum_{k=2}^{N_2} \hat{u}_{jk}$.

This approach allows for the efficient computation of $\hat{u}_{1k}$ through similar reasoning. By employing FFT and its inverse (iFFT), the algorithm accelerates the matrix approximation process during both compression and decompression phases, with a computational complexity of $O(N^2 \log N)$. This Fourier-based approach significantly enhances computational efficiency compared to traditional matrix computation methods, thereby ensuring the throughput of compression and decompression operations while maintaining lossless reconstruction of the original data.

## 3.6 High-Order Mechanisms

While the proposed 9-point difference operator $\mathcal{L}$ is primarily designed to capture second-order PDE dynamics, many real-world datasets may exhibit higher-order effects arising from more complex governing equations or fine-scale features. These higher-order mechanisms can manifest in the source term $f$, potentially limiting the compression performance if not properly addressed.

To better characterize these higher-order properties, a preprocessing step involving the introduction of diffusive operators can be employed. Let $\mathcal{D}$ denote a diffusive operator, such as $\mathcal{D} = \alpha \Delta^2 + \beta \Delta$, where $\Delta$ is the Laplacian and $\alpha, \beta$ are constants. We can apply this operator to the original data $u$ to obtain a smoothed version $\tilde{u} = \mathcal{D}u$, which effectively filters out the fine-scale, high-order features.

The compression is then performed on $\tilde{u}$ instead of $u$, yielding a sparse source term $\tilde{f}$ that encapsulates the higher-order effects $\mathcal{L}\tilde{u} = \tilde{f}$. During decompression, the inverse diffusive operator $\mathcal{D}^{-1}$ is applied to recover the original data

$$u = \mathcal{D}^{-1}(\tilde{u}) = \mathcal{D}^{-1}(\mathcal{L}^{-1}\tilde{f} + \tilde{u}_0). \tag{3}$$

This preprocessing strategy allows capturing high-order mechanisms by absorbing them into the sparse source term $\tilde{f}$, while the 9-point operator $\mathcal{L}$ focuses on representing the underlying second-order dynamics in the smoothed data $\tilde{u}$. The diffusive operator approach maintains computational efficiency, lossless compression/decompression, and enhances handling of complex, high-order data. The diffusive operator $\mathcal{D}$ can be tailored to the dataset, potentially with learnable parameters for optimized filtering. Iterative applications of $\mathcal{D}$ enable progressive extraction of high-order features at multiple levels.

## 4 Algorithm

The algorithm uses an optimized 9-point template to achieve lossless compression and decompression of data, which is mainly divided into two parts: compression and decompression.

---

**Algorithm 1** Compression based on optimizing source term

---

**Require:** Data $u$ and precision $m$
**Ensure:** Coefficients $C$, source term $f$, and boundary values $u^{bd}$
 1: Select initial template $\mathcal{L}_0 u$ (hyperbolic, elliptic, or other) and coefficient truncation precision $n$.
 2: Set $S1 \leftarrow 1$, the adjustment count $S2 \leftarrow 1$, and the upper limit for the adjustment count $S2^*$.
 3: **while** $S1 > 0$ and $S2 < S2^*$ **do**
 4:     **if** high-order mode is required **then**
 5:         Compute the source term $u_1 \leftarrow \mathcal{L}_0 u$ using the initial template.
 6:     **else**
 7:         $u_1 \leftarrow u$.
 8:     **end if**
 9:     Optimize the template coefficients $C \leftarrow \arg\min_C \|\mathcal{L}_c u_1\|$.
10:     Obtain the source term $f \leftarrow \mathcal{L}_c u_1$.
11:     Truncate the source term to $10^{-(m+n)}$ (or binary truncation): $f \leftarrow$ truncated source term.
12:     Encode $f$, increment $S2$, check the residual error, and verify the compression rate.
13:     **if** the compression rate is unsatisfactory **then**
14:         Adjust $n$, set $S1 \leftarrow 1$.
15:     **else**
16:         $S1 \leftarrow 0$.
17:     **end if**
18: **end while**

---

The compression part includes selecting the initial template, setting the coefficient truncation precision, and optimizing the template coefficients to minimize the residual error and information entropy of the compressed data. The compression process also includes truncating the source term to the required precision and encoding it using an appropriate encoding scheme. The compression part terminates when the compression rate is satisfactory or when the maximum number of iterations is reached.

---

**Algorithm 2** Decompression based on optimized source term

---

**Require:** Template coefficients $C$ (preprocessing template $C_0$, if any), source term $f$, boundary values $u^{bd}$ (preprocessing boundary $u_0^{bd}$, if any), and data precision $m$.
**Ensure:** Data array $u$.
 1: **if** high-order mode is required **then**
 2:     Assemble the stiffness matrix $K_0(C_0)$.
 3:     Assemble the right-hand side $d(u_0^{bd}, f)$.
 4:     Solve the equation $K_0 f_1 = d$.
 5: **else**
 6:     $f_1 \leftarrow f$.
 7: **end if**
 8: Assemble the stiffness matrix $K(C)$.
 9: Assemble the right-hand side $d(u^{bd}, f_1)$.
10: Solve the equation $Ku = d$.
11: Reconstruct $u$ with the precision of original data.

---

The decompression part includes assembling the stiffness matrix and right-hand vector using the optimized template coefficients, solving the linear system to obtain the compressed data, and truncating it to the original precision. The decompression process also includes decoding the compressed source term using the same encoding scheme and using it to reconstruct the original data.

## 5    Experimental Results

This section presents the experimental results of applying our proposed lossless compression algorithm to the CESM-ATM and Hurricane datasets from the SDRBench[18]. The performance is evaluated based on the original data, transformed data (source terms post-transformation), reconstruction error, frequency distribution plots, compression ratio, throughput, and comparison with existing algorithms such as Zstandard (zstd)[5] and fpzip[3]. All tests were conducted on a Mac with M1 Silicon, macOS 14.1.2, 16GB RAM.

### 5.1    Compression and Reconstruction

The original datasets were first transformed using our proposed method to generate source terms suitable for encoding. The transformed data were then compressed, and the reconstruction error was calculated by comparing the original data to the data reconstructed from the compressed representation.

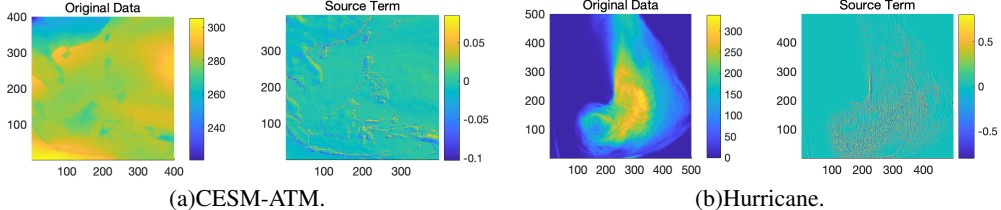

(a)CESM-ATM.                                    (b)Hurricane.

Figure 5: Demo of the proposed scheme on CESM-ATM and Hurricane datasets.

Figure 5 illustrates the results obtained by applying the proposed scheme to the CESM-ATM and Hurricane datasets. For the CESM-ATM data, while the original data values span a range of 220 to 300, the source term exhibits a more compressed range of -0.1 to 0.1, indicating effective compression. Notably, the reconstruction error was found to be around $10^{-11}$, which is smaller than the least significant digit of single-precision floating-point representation. For the Hurricane dataset, the reconstruction error is even smaller, on the order of $10^{-12}$. These results demonstrate that the proposed method preserves the numerical precision essential for scientific computations. The extremely low reconstruction errors ensure that the compressed data can be used reliably in high-precision applications, maintaining the integrity of the original datasets for subsequent analyses and simulations.

### 5.2    Frequency Distribution Analysis

To further illustrate the effectiveness of our compression method, we compared the frequency distribution plots of the data before and after compression. The left represents the original data, while the right represents the stored source terms. The histograms employ logarithmic binning for clarity, demonstrating the reduced mean and standard deviation of the source term, which contributes to decreased entropy and, consequently, a better compression rate.

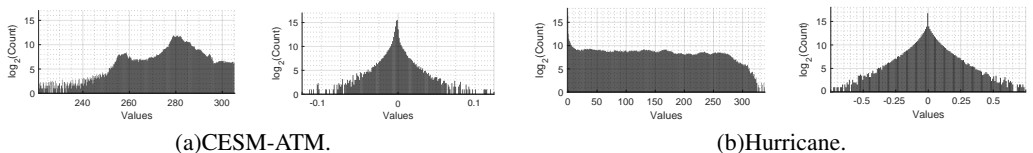

(a)CESM-ATM.                                    (b)Hurricane.

Figure 6: Frequency distribution plots of data before and after compression.

## 5.3 Performance Evaluation

We evaluate MeLLoC against several state-of-the-art compression algorithms using two key metrics: compression ratio and throughput. The compression ratio indicates the efficiency of data reduction, with higher values representing more compact representations. Throughput measures the speed of compression and decompression processes, which is crucial for handling large scientific datasets. Table 1 presents a comprehensive comparison across CESM-ATM and Hurricane datasets.

Table 1: Performance comparison.

| | CESM-ATM | | | | Hurricane | | |
| Method | Ratio | Compression | Decompression | Method | Ratio | Compression | Decompression |
| --- | --- | --- | --- | --- | --- | --- | --- |
| ALP | 1.16 × | 46.93 Mb/s | 1054.95 Mb/s | ALP | 1.11 × | 45.74 Mb/s | 973.63 Mb/s |
| FPZIP | 1.63 × | 59.68 Mb/s | 70.94Mb/s | FPZIP | 1.63 × | 41.22Mb/s | 53.95Mb/s |
| ZFP | 1.02× | 96.17 Mb/s | 81.97 Mb/s | ZFP | 1.01 × | 102.95 Mb/s | 68.06 Mb/s |
| Blosc | 1.30× | 293.71 Mb/s | 632.76 Mb/s | Blosc | 1.12 × | 888.65 Mb/s | 6516.29 Mb/s |
| Gzip | 1.89 × | 1.40 Mb/s | 266.94 Mb/s | Gzip | 1.00 × | 33.25 Mb/s | 212.35 Mb/s |
| Zstandard | 2.69 × | 105.51Mb/s | 152.81Mb/s | Zstandard | 2.78 × | 69.51Mb/s | 271.32Mb/s |
| **MeLLoC** | **3.36 ×** | **188.77** Mb/s | **179.76**Mb/s | **MeLLoC** | **3.29×** | **206.80**Mb/s | **190.35**Mb/s |

ALP and FPZIP are lossless floating-point compression algorithms, while ZFP is set to lossless mode with zero error tolerance. Blosc, Gzip, and Zstandard are general-purpose lossless compression algorithms. MeLLoC, the proposed method, consistently achieves the highest compression ratios (3.36× for CESM-ATM and 3.29× for Hurricane) while maintaining competitive throughput.

## 5.4 Scalability Performance Analysis

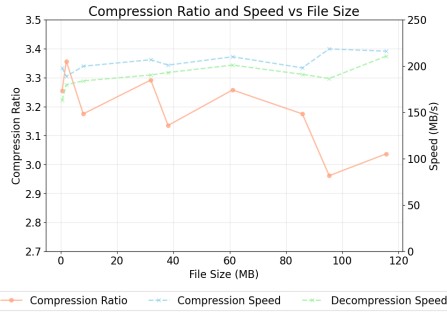

Figure 7: Performance Metrics Across File Sizes.

In Figure 7, the compression ratio, represented by the orange line, shows a slight decline as file size increases, dropping from around 3.4 to just above 3.1 for the largest files. Compression and decompression speeds, depicted by the blue and green dashed lines respectively, remain relatively stable across varying file sizes, with minimal fluctuations. Compression speeds consistently outperform decompression speeds, both hovering within a range of 150 to 200 MB/s across all file sizes. This indicates a stable and efficient scalability pattern in the data compression process.

## 6 Conclusion

In this paper, we present a novel mechanism-learning method for compressing scientific data. Our proposed scheme effectively preserves the true precision of the data while enabling subsequent lossless compression. Additionally, the approach offers flexibility by allowing the use of various lossless encoding algorithms, which can be selected based on factors such as availability, speed, and resulting file sizes, tailored to specific application requirements. Experiments demonstrate that the proposed method has competitive reconstruction performance with the general-purpose methods, while providing a novel perspective on scientific data compression. MeLLoC has several limitations. As discussed in Section 3.1, its performance is contingent upon the accuracy of the differential equation models representing the data. The method is particularly suitable for scientific data whose order of magnitude aligns well with the assumed physical models. For datasets that do not fit these criteria, the method may not be as effective. The precision control, crucial for optimizing compression, also necessitates careful calibration, which might be challenging with diverse data characteristics. Despite these constraints, MeLLoC presents a valuable approach for scientific data compression. In future work, we intend to apply the 'mechanism-learning' concept to several pertinent real-world domains, such as medical imaging, oceanography, and other related fields.

## Acknowledgments and Disclosure of Funding

This work is supported by National Key Research and Development Programs of China (No. 2023YFA1009103), National Natural Science Foundation of China (Nos. 12201386, 12241103), Science and Technology Commission of Shanghai Municipality (23JC1400501) and the Sino-German Mobility Programme (M-0187) by Sino-German Center for Research Promotion.

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
