# OpenReview forum: "MeLLoC: Lossless Compression with High-order Mechanism Learning"
_NeurIPS.cc/2024/Conference — NeurIPS 2024 poster_

### Official Review · Reviewer_5CqK · 2024-07-07

**Soundness:** 3
**Presentation:** 3
**Contribution:** 3
**Rating:** 8
**Confidence:** 4

**Summary:**

This paper introduces a novel approach combining high-order mechanism learning with classical encoding techniques to enhance lossless compression for large-scale scientific floating-point data. The core innovation lies in treating data as discrete samples from a physical field governed by differential equations and solving inverse problems to identify compressible coefficients. Experiments demonstrate MeLLoC's superior performance over existing methods, achieving better compression ratios and computational efficiency.

**Strengths:**

1) Innovative Approach: Combines mechanism learning with classical encoding, leveraging differential equations to compress scientific data effectively.
2) Superior Performance: Outperforms state-of-the-art lossless compression techniques in terms of compression ratios and computational efficiency.
3) Comprehensive Experiments: Extensive testing on various datasets highlights the robustness and efficacy of the proposed method.
4) Flexibility: Capable of handling different types of scientific data, including those with high-order information and noise.

**Weaknesses:**

1) Model Dependency: Performance relies heavily on the accuracy of the differential equation models representing the data.
2) Complex Calibration: Precision control requires careful calibration, which can be challenging with diverse data characteristics.
3) Computational Intensity: Despite improved efficiency, the method still demands significant computational resources for large datasets.
4) Generalization: May not be as effective for datasets that do not conform well to the assumed physical models.

**Questions:**

1) Clarification on Differential Equation Models:
Can you provide more details on how you determine the appropriate differential equation models for different datasets?
How sensitive is the performance of MeLLoC to the choice of these models?
2) Precision Control:
Could you elaborate on the process for calibrating the precision control?
Are there any guidelines or heuristics you follow to balance the compression efficiency and computational cost?

**Limitations:**

The authors have recognized and addressed several limitations of their work:
1) Model Dependency: The performance of MeLLoC depends on the accuracy of the differential equation models representing the data.
2) Calibration Complexity: Precision control requires careful calibration, which can be challenging with diverse data characteristics.
3) Computational Intensity: The method demands significant computational resources for large datasets.
4) Generalization Issues: MeLLoC may not be as effective for datasets that do not align well with the assumed physical models.

---

> ### Author Rebuttal · Authors · 2024-08-06
>
> We thank the reviewer for the overall positive feedback and the valuable comments. We have revised our manuscript taking your concerns and suggestions into consideration.
>
> To answer your questions/comments:
> > Q1: Clarification on Differential Equation Models: Can you provide more details on how you determine the appropriate differential equation models for different datasets? How sensitive is the performance of MeLLoC to the choice of these models?
>
> Thanks for your valuable comments. In our work, mechanism learning aims to discover the differential equation model that best describes the data. The identification of the mechanism is explained in the "Model identification and its well-posedness" section of the "Author Rebuttal". Please kindly refer to that. Through optimization, we obtain the optimal parameters $\theta^*$ in Figure 1. These optimal parameters directly correspond to the coefficients in the second-order PDEs.
>
> We also appreciate the reviewer's observation on sensitivity. Taking the CESM-ATM dataset as an example, when using fixed templates as shown in Figure 2(c), we observe the following compression ratios:
> 1) Laplacian template: $2.67\times$;
>
> 2) Hyperbolic template: $3.29\times$;
>
> 3) Parabolic template: $1.45\times$.
>
> As shown by varying compression ratios, the performance is therefore sensitive to model choice. By MeLLoC, the model is optimized and the learned template approach consistently outperforms fixed templates, achieving 3.36x as shown in Table 1.
>
> It also suggests that the underlying physical mechanisms in the CESM-ATM dataset align more closely with Hyperbolic Equations/Transportation Equations.  Such insights could potentially inform future modeling strategies and enhance our understanding of atmospheric dynamics in climate models.
>
> > Q2: Precision Control: Could you elaborate on the process for calibrating the precision control? Are there any guidelines or heuristics you follow to balance the compression efficiency and computational cost?
>
>  We appreciate your question. Our method ensures lossless compression while optimizing computational efficiency through the following process:
> 1) We observe that the source term $f = L(u) = \sum_{i=1}^9 C_i u_i$ is computed with precision $10^{-(m+n)}$, where $m$ is the original data precision and $n$ is the model coefficient precision. We can maintain lossless while the solver for $K_\mathcal{L}u^{in} = b_{u_{bd},f}$ has precision capacity of $10^{-(m+n)}$ (this is feasible with proper $n$). As $n$ increases, the admissible set for $C_i$ enlarges, contributing to a lower absolute value for $f$ but higher precision. The best compression ratio is reached when significant digits of $f$ reach a minimum.
> 2) We optimize $n$ for coefficients $C_i$ by starting with high precision and gradually reducing it while monitoring reconstruction error and compression ratio. With several calibrations, $n$ can be fixed for the remaining dataset if the compression ratio for the following batches shows no significant fluctuation.
> 3) The approach considers different scenarios to balance significant digits and value magnitudes, as shown in Figure 3(b).
>
> MeLLoC optimizes compression efficiency within lossless constraints, balancing perfect reconstruction with computational feasibility. The precision control is adaptive and can be tailored to different scientific datasets.

---

> > ### Comment · Reviewer_5CqK · 2024-08-12
> >
> > Thank you to the author for carefully responding to my question.
> > I think the method proposed in this paper achieves excellent compression performance and contributes to the field of compression. I will keep my score.

---

### Official Review · Reviewer_VS2J · 2024-07-08

**Soundness:** 2
**Presentation:** 2
**Contribution:** 3
**Rating:** 4
**Confidence:** 3

**Summary:**

This paper introduces MeLLoC (Mechanism Learning for Lossless Compression), an approach that combines high-order mechanism learning with classical encoding to enhance lossless compression for scientific data. The core concept is to interpret the data as discrete samples derived from an underlying physical field described by differential equations. It addresses an inverse problem to identify coefficients of the governing equations, aiming to achieve a more compressible numerical representation.

**Strengths:**

The proposed MeLLoC is innovative in its approach of treating data as samples from a discretized physical field. It solves an inverse problem to determine the source terms of the governing differential equations, resulting in a more compressible numerical distribution. MeLLoC demonstrates a higher compression ratio and faster compression speed compared with existing methods.

**Weaknesses:**

The readability is weak. More detailed explanations and experimental analyses are necessary. For instance, the concept of Mechanism Learning is not novel and should be introduced in more detail. Additionally, the comparison with other methods and datasets is insufficient. For example, the authors should compare the proposed approach with [1], [2], and other relevant methods, if feasible.

Ref:

[1] Knorr, Fabian, Peter Thoman, and Thomas Fahringer. "ndzip: A high-throughput parallel lossless compressor for scientific data." 2021 Data Compression Conference (DCC). IEEE, 2021.

[2] Afroozeh, Azim, Leonardo X. Kuffo, and Peter Boncz. "ALP: Adaptive Lossless floating-Point Compression." Proceedings of the ACM on Management of Data 1.4 (2023): 1-26.

**Questions:**

1.	Is [1] a lossless compression method? It is recommended to clarify the differences between [1] and this paper for a better understanding.
2.	Hurricane appears to have a smaller reconstruction error as in Figure 5. Why then does Hurricane exhibit a lower compression ratio compared to CESM-ATM?
3.	What are the challenges in applying this method to other domains (e.g., medical imaging, oceanography as mentioned in the Conclusion)?

Ref: Luo, Xinyue, et al. "Precision-preserving Compression of Scientific Data: Learn Mechanism from Data." 2024 Data Compression Conference (DCC). IEEE, 2024.

---

> ### Author Rebuttal · Authors · 2024-08-06
>
> We would like to thank the reviewer for the thoughtful comments and valuable suggestions to make the paper clearer. We have revised our manuscript taking all comments and suggestions into consideration.
> > Q1: Is [1] a lossless compression method? It is recommended to clarify the differences between [1] and this paper for a better understanding.
>
> Thank you for your careful reading and instructive comments. While [1] does not achieve true lossless compression, our approach is designed specifically to ensure perfect reconstruction of the original data. In the revised version, we have emphasized the differences from [1] in terms of model principles and specific implementation, and we have added a more detailed analysis in the Appendix on the lossless implementation and the principles of fast solver. A discussion can also be found in the "Differences between [Luo et al.(2024a)] and our work" section of the "Author Rebuttal".
>
> > Q2: Hurricane appears to have a smaller reconstruction error as in Figure 5. Why then does Hurricane exhibit a lower compression ratio compared to CESM-ATM?
>
> Thank you for this insightful question. The proposed algorithm achieves lossless compression on both CESM-ATM and Hurricane datasets since the reconstruction error is less than $10^{-7}$. The compression ratio is not directly related to reconstruction error. The ratio depends on two main factors:
> (a) The statistical properties of the source term $f$ after transformation.
> (b)The noise level of the original data. A sparser $f$ leads to better compression. Additionally, data with lower noise levels also leads to better compression.
> From this perspective, we can infer that CESM-ATM likely yields a more compressible $f$ than Hurricane data in terms of PDE representation, or that the CESM-ATM data contains lower levels of noise.
>
> > Q3: What are the challenges in applying this method to other domains (e.g., medical imaging, oceanography as mentioned in the Conclusion)?
>
> We appreciate your valuable question. Our method has extended to medical imaging and oceanography, presenting both opportunities and challenges. In medical imaging analysis, the identification of high-order information (source term $f$) enables us to extract pathological features from images, such as detecting anomalies in OCTA scans. This capability arises from distinguishing between normal tissue mechanisms and abnormal external influences. However, interpreting these results requires expert guidance to correlate specific high-order information patterns with particular diseases. In oceanography, mechanism identification also plays a crucial role. By learning PDEs(perhaps higher-order ones), we can identify governing equations, which help us determine the impacts of diffusion and convection terms. Based on the identified models, we could predict the future evolution of oceanic systems. The main challenge here lies in the need for continuous observation to achieve accurate model characterization. For instance, robust ocean mechanism identification often requires data assimilation techniques.
>
> **Comparative Analysis with Other Methods**
>
> Thank you for your suggestion to include comparisons with additional relevant methods. We carefully read the papers you provided and made the following adjustments to our study: We did not include ndzip in our comparative experiments because it is also based on the Lorenzo predictor, which is similar to FPZIP and it requires a parallel computing environment.
> We have supplemented our study with comparative experiments involving:
> 1) Floating-point compression algorithms: ALP and ZFP;
> 2) General-purpose lossless compression algorithms: Blosc and Gzip.
>
> The detailed results of these additional experiments can be found in the "Additional Experiment Results" section of the "Author Rebuttal".
>
> **More explanation on Mechanism Learning**
>
> Thanks for your invaluable comment. We have provided a more comprehensive introduction to mechanism learning in the revised manuscript.

---

> > ### Comment · Reviewer_VS2J · 2024-08-13
> >
> > Thanks for your rebuttal. My rating will be still close to a Borderline.

---

### Official Review · Reviewer_XGH7 · 2024-07-13

**Soundness:** 2
**Presentation:** 2
**Contribution:** 1
**Rating:** 3
**Confidence:** 4

**Summary:**

This paper propose a near lossless compression method named MeLLoC to compress the scientific data by learning the inherent mechanisms. By solving the inverse problem of Partial Differential Equations (PDEs), MeLLoC transforms the scientific data from original data domain into discretized source domain, which is much easier to compress. Besides, several techniques including precision control, fast Fourier-based solver and preprocessing for high-order mechanisms are proposed to facilitate the compression efficiency. Experimental results show that MeLLoC outperfoms two previous methods (FPZIP and Zstandard).

**Strengths:**

The paper studies an interesting and promising direction that leverages mechanism learning to compress scientific data.

**Weaknesses:**

1. The proposed MeLLoC is built on the previous work [Luo et al. (2024a)]. However, it seems that the whole architecture of MeLLoC is the same as that in [Luo et al. (2024a)] and there is no new contribution.

2. The proposed method is not truly lossless compression as claimed, but is near-lossless compression. Section 5.1 shows that the largest reconstruction error between the original data and reconstructed data is approximately in the order of 10^{-11}. However, A single-precision floating-point number can be accurate to the order of 10^{-38}.

3. Presentation in Section 3 is somewhat confusing. For example, it is not clear why the difference operator $\mathcal{L}$ is formulated in exactly a 9-point form and how the loss function $F$ is minimized.

4. Evaluations in Section 5.3 are not sufficient. Some recent methods such as [Klöwer et al.(2021)] and [Luo et al.(2024a)] are not compared.

5. Sections 2.1 and 2.2 are not related to the topic of this paper. It would be better to provide some background about PDEs and mechanisms of scientific data to inform more preliminary knowledge to Section 3.

**Questions:**

1. What is the differences between MeLLoC and the method proposed in [Luo et al. (2024a)]?

2. Does MeLLoC achieve lossless compression or near-lossless compression? If it is near-lossless compression, what is the actual bit-rate to achieve lossless compression?

3. Please explain the formulation of difference operator $\mathcal{L}$ and loss function $F$.

4. What is the performance of MeLLoC compared to [Klöwer et al.(2021)] and [Luo et al.(2024a)]?

**Limitations:**

The authors have discussed the limitations.

---

> ### Author Rebuttal · Authors · 2024-08-06
>
> We greatly appreciate the careful review of the manuscript. We sincerely hope that the following answers will better illustrate our work. We also recommend that the reviewer read the “Author Rebuttal.” We hope the reviewer finds the efforts and improvements we made in both the theoretical and experimental aspects.
>
> ### 1. Clarification on Lossless Properties (To answer Q1 and Q2)
>
> We appreciate your observation regarding the precision of single-precision floating-point numbers. However, it appears there is a misunderstanding. A single-precision float indeed has about 7 decimal digits of precision, which corresponds to approximately $10^{-7}$. MeLLoC achieves true lossless compression, allowing for perfect reconstruction of the original data from its compressed form. By controlling the precision of coefficients and source terms based on the inherent properties of the scientific data, this approach allows us to achieve better compression ratios while maintaining lossless compression.
> In contrast, the work by [Luo et al. (2024a)] which incorporates a noise separation process, is indeed near-lossless. We emphasize that the proposed method does not introduce any loss of original information, and we have clarified this point in the revised manuscript.
>
> ### 2. Novelty: Lossless Compressor & Fast Solver (To answer Q1)
>
> Thanks for your valuable comments. We would like to highlight the novelty of MeLLoC: its lossless properties and a fast solver for the compression and decompression processes. As demonstrated above, the proposed method achieves lossless compression. Another novel aspect is the introduction of a fast solver based on PDE theory.  Our fast solver significantly accelerates both the compression and decompression processes. Specifically, our method has substantially enhanced performance, improving the compression throughput by 839.47\% and the decompression throughput by 352.27\% compared to [Luo et al. (2024a)].
>
> Unlike traditional compression methods, MeLLoC transforms the data into a sparser representation, i.e., high-order mechanisms derived from PDE models. The compression is accelerated due to the linearity and sparsity of this local representation, as shown in Figure 2. This allows for direct optimization to find the extremum. The existence and uniqueness of the minimizer during optimization are guaranteed by PDE theory. For more details, please refer to the "Model identification and its well-posedness" section in the "Author Rebuttal." This is the first application of such rapid computational methods in the field of data compression. We believe this contribution is substantial and distinct from previous works, including [Klöwer et al.(2021)] and [Luo et al. (2024a)].
>
> ### 3. Explanation of Technical Details (To answer Q3)
> Thanks for your comments. We would like to expand on Section 3.1 to further explain how our method interprets data and how we formulate $\mathcal{L}$ and $F$. The revised paper includes more detailed illustrations and analysis of model identification and the well-posedness of identification in the Appendix. For details, please kindly refer to the "Model identification and its well-posedness" section in the "Author Rebuttal".
>
> ### 4. Evaluation Against Recent Methods (To answer Q4)
> We appreciate your suggestion to include comparisons with recent methods.
> In response, we have expanded our evaluation in Section 5.3 to include comparisons with other state-of-the-art lossless algorithms, ALP[1], ZFP[2], Blosc, and Gzip, as suggested by Reviewer VS2J. Please find the result in the attached PDF file in the "Author Rebuttal" section.
>
> > References:
> >
> > [1] Afroozeh, A., Kuffo, L. X., \& Boncz, P. (2023). ALP: Adaptive Lossless Floating-Point Compression. Proceedings of the ACM on Management of Data, 1(4), 1-26.
> >
> > [2] Lindstrom, P. (2014). Fixed-rate compressed floating-point arrays. IEEE Transactions on Visualization and Computer Graphics, 20(12), 2674-2683.
>
> ### 5. Relevance of Sections 2.1 and 2.2 (To respond to Weakness 5)
>
> We apologize for the confusion caused by Sections 2.1 and 2.2. We intended to convey that the use of partial differential equations (PDEs) to describe human vision and observable phenomena throughout scientific history represents a process of compression and decompression within human intelligence. The process of abstracting observable phenomena, specifically translating human vision into mathematical equations, is a form of information extraction. This mechanism-learning process inspired us to propose a new data compression method, where compression and decompression reflect the extraction of data features and the intelligent associations behind human vision related to data mechanisms. We revised these sections to provide more relevant background on PDEs and the mechanisms of scientific data, ensuring that they better inform the preliminary knowledge necessary for understanding Section 3.
>
> In summary, we appreciate your insights and have made revisions carefully according to your comments, to clarify the contributions and technical details of MeLLoC. Thank you again for your valuable feedback.

---

> > ### Comment · Reviewer_XGH7 · 2024-08-13
> >
> > I would like to thank the authors for their efforts in preparing the rebuttal. However, I still have concerns on the claim that the proposed MeLLoC achieves lossless compression. According to IEEE 754, single-precision floating-point uses 32 bits in total, consisting of three parts: sign bit (1 bit), exponent (8 bits), and mantissa (23 bits). It can cover a vast range of values (from around $10^{-38}$ to $10^{38}$). I recommend the authors to provide multiple concrete examples that cover different magnitudes of single-precision floating-point numbers (e.g., with more than 7 decimal digits) to demonstrate that the proposed method can perfectly reconstruct these 32-bit single-precision floating-point numbers. Since the authors claim that the lossless property is the main novelty of the proposed MeLLoC compared to previous work (Luo et al. 2024a), I will maintain my rating before this issue is clarified.

---

> ### Author Response · Authors · 2024-08-13
> **Author response about the single-precision issue**
>
> Thanks very much for the reviewer’s kind reply. We clarify that the capacity of significant numbers for single-precision floating numbers is about 7 decimal digits, while the values cover $-10^{-38}$ to $10^{38}$. The example considered in the manuscript has values spanning from 180-300, which have the same order of magnitude. Therefore, we use error level $10^{-7}$ to display the lossless effect visually. It is common for scientific data, like atmospheric and oceanic remote sensing data, to have close order spans even fixed absolute precision. Of course, there might be cases with large order spans among the data. In that case the ‘outliers’ will be detected in our method and maintained lossless by storing the residuals separately, the compression rate will be affected. The method performs best for the above mechanism data, while in extreme cases with random noise, there is little mechanism and the method will no longer work. We are sorry for not explaining this clearly in the previous communications. Thanks again for your effort in reviewing our work.

---

### Author Rebuttal · Authors · 2024-08-06

We thank all the reviewers for their insightful comments and suggestions. Accordingly, we try our best to make substantial revisions. The revised article now contains the extensions of the proposed lossless compression framework, and the additional experiments on comparison studies. We have addressed the reviewers' questions in our respective responses.

### Additional Experiment Results{Reviewer XGH7,Reviewer VS2J}
During the rebuttal period, we have performed additional experiments as shown in Table 1 of the attached PDF, which will be the updated version of Table 1 of the main manuscript. The updated results demonstrate a comprehensive comparison between MeLLoC and several state-of-the-art lossless compression algorithms, including ALP, FPZIP, and ZFP, which are specifically designed for floating-point number compression. MeLLoC significantly outperforms general-purpose algorithms in terms of compression ratio. While its throughput may not match that of ALP or Blosc, it introduces a novel compression paradigm for scientific floating-point data. Figure 1 in the attached PDF visualizes these results, with the second graph showing Combined Performance (compression rate * log(speed)). MeLLoC demonstrates a clear advantage over other algorithms in this balanced metric, balancing high compression ratios with reasonable processing speeds.

### Differences between [Luo et al. (2024a)] and our work{Reviewer XGH7,Reviewer VS2J}
The method proposed by [Luo et al. (2024a)] is a near-lossless compression technique, as it incorporates noise separation. Consequently, it cannot achieve perfect reconstruction within the digit precision of the original data. In contrast, MeLLoC is a true lossless compressor. To achieve industry-standard high throughput, we have optimized MeLLoC's computational process based on second-order PDE theory. The well-posedness from PDE theory also ensures the feasibility of high-order data representation.

### Model identification and its well-posedness{Reviewer XGH7,Reviewer 5CqK}
Consider the underlying model of the data. For $u \in C^4(\Omega)$, $\Omega \subset \mathbb{R}^2 $,  the nine-point difference template is related to the second-order linear differential operator as
$$\sum_{k, l=-1}^1 C_{k, l} u(x+k h, y+l h) =  [2(c_1+c_5-c_6) h \partial_x+2(c_2+c_5+c_6) h \partial_y  +(c_3+\frac{1}{2} c_7+\frac{1}{2} c_8) h^2 \partial_{xx}^2+(c_4+\frac{1}{2} c_7+\frac{1}{2} c_8) h^2 \partial_{yy}^2 +(c_7-c_8) h^2 \partial_{xy}^2+c_9] u(x, y)+o(h^2).$$
where $C_{k,l}$ are the coefficients in Figure 2(a), the subscripts ${k,l}$ represent the relative position to the data point $(x,y)$.
The relationship between $C_{k,l}$ and $c_n$ can be represented as
$$\mathbf{C}=c_1 \mathbf{A}_1+c_2 \mathbf{A}_2+c_3 \mathbf{A}_3+c_4 \mathbf{A}_4+c_5 \mathbf{A}_5+c_6 \mathbf{A}_6+c_7 \mathbf{A}_7+c_8 \mathbf{A}_8+c_9 \mathbf{A}_9,$$

where $\mathbf{C}$ is the matrix of $C_{k,l}$, $\mathbf{A}_n$ are basis matrices, and $c_n$ are corresponding coefficients.
$\{\mathbf{A}_n\}$ are defined as
$$
\mathbf{A}_1 = [0,0,0;-1,0,1;0,0,0], \mathbf{A}_2 = [0,1,0;0,0,0;0,0,-1],
\mathbf{A}_3 = [0,0,0;1,-2,1;0,0,0], $$
$$ \mathbf{A}_4 = [0,1,0;0,-2,0;0,1,0], \mathbf{A}_5 = [0,0,1;0,0,0;-1,0,0],  \mathbf{A}_6 = [1,0,0;0,0,0;0,0,-1], $$
$$
\mathbf{A}_7 = [0,0,1;0,-2,0;1,0,0],  \mathbf{A}_8 = [1,0,0;0,-2,0;0,0,1],
\mathbf{A}_9 = [0,0,0;0,1,0;0,0,0].$$

Based on this representation, encoding $u$ becomes encoding the sparser high-order term $o(h^2)$, i.e., the source term $f$. Therefore, the optimization objective is to obtain a minimized high-order term, which can be mathematically expressed as
$$\{C_{k,l}\}^{*} = argmin_{C_{k,l}} F({C_{k,l}};u) = argmin_{C_{k,l}} (\sum_{i, j}\sum_{k, l=-1}^1 C_{k, l} u(i+k h, j+l h))^2.$$

Once the template $\theta:=C_{k, l}$ is learned, one can calculate the coefficients of the differential operator, allowing one to classify the mechanism as elliptic, parabolic, or hyperbolic due to the reversibility of $C_{k, l}$ and $c_n$.

Next, we will briefly explain the solvability of the model identification problem. The well-posedness of the compression process is considered by the following formulation. The above minimization problem is equivalent to solving the least square problem: $$Ac = 0,$$

where $c = [C_1, \cdots, C_9]^T$, $A \in \mathbb{R}^{N\times 9}$, $N$ is the number of data points in domain $D$, $\mathcal{P}: D \to \mathbb{R}$, $k = \mathcal{P}(i,j)$, $(i,j) \in D$ is the index after rearranging the data into a one-dimensional vector, $(i,j) = \mathcal{P}^{-1}(k)$, $(k = 1, \cdots, N)$ is its inverse mapping.
$$A_{k,\cdot} = [u_{i-1,j-1}, \cdots, u_{i+1,j+1}], \quad (i,j) = \mathcal{P}^{-1}(k).$$

Therefore, to find non-trivial solutions is to obtain the null space (kernel) of $B = (A^TA)$. There is only a trivial solution if $B$ is rank full while otherwise there is no uniqueness. To address this issue, if we set the coefficient of $u_{i,j}$ to -1 (i.e., set $C_5$ to -1 in Figure 2 (a)), and fix the template size to 8, then the above problem becomes:
$$\tilde{A}\tilde{c} = b,$$

where $\tilde{c} \in \mathbb{R}^8$, $\tilde{A} \in \mathbb{R}^{N\times 8}$,
and $b = [u_{\mathcal{P}^{-1}(1)}, \cdots, u_{\mathcal{P}^{-1}(N)}]^T \in \mathbb{R}^N$. Then, the above problem has a unique least squares solution $\tilde{c} = \tilde{A}^\dagger b$, provided the data are not all degenerated. We assemble $\tilde{A}$ and directly solve the pseudo inverse, which serves as our fast solver for the compression process.

Finally, we thank all the reviewers again for your insightful comments. We do believe that the revised article is much improved not only from the theoretical aspect but also from the experimental aspect. We are happy to answer any further questions the reviewer might have.

---

### Decision · Program_Chairs · 2024-09-25

**Decision:**

Accept (poster)

**Comment:**

The reviewers had various questions and comments mainly related to differentiation relative to previous work by Luo 2024a as well as whether the target precision can be stated as lossless. The authors addressed every point raised. The authors also provided more experimental results as requested. Two reviewers, who took negative stances, were reluctant to change scores. One of these two reviewers had no counter argument. The other reviewer seemed to have only one issue left that had to do with the definition of the "lossless" property. Overall, the author rebuttals adequately addressed concerns raised. While likely to be effective for only certain types of data, the proposed method appears new and interesting.